# ALO: Addressing Class Imbalance in Radiology Report Generation through Anatomy-Level Oversampling

**Lukas Buess**[*1]                Lukas.Buess@fau.de
**Robert Kurin**[*1]              robert.kurin@studium.fau.de
**Adarsh Bhandary Panambur**[1]      Adarsh.Bhandary.Panambur@fau.de
**Tomas Arias-Vergara**[1]            Tomas.Arias@fau.de
**Andreas Maier**[1]              Andreas.Maier@fau.de
[1] *Pattern Recognition Lab, Friedrich-Alexander-Universität Erlangen-Nürnberg, Erlangen, Germany*

**Editors:** Accepted for publication at MIDL 2026

## Abstract

Radiology report generation aims to connect visual understanding with clinical language, yet most methods rely on free-text supervision, which is highly variable and difficult to evaluate. Clinical datasets are also dominated by normal findings, causing models to underreport abnormalities. While recent works focus on architectural advances, we show that structured supervision and balanced sampling can yield substantial gains in clinical performance. We convert free-text reports into structured anatomy-level representations and introduce Anatomy-Level Oversampling (ALO), a data centered sampling strategy that balances normal and abnormal findings for each anatomical region. This structure provides consistent supervision and enables more informative evaluation. Across three public datasets, ALO improves sensitivity to pathological findings while remaining fully model agnostic. On internal validation, ALO increases F1-Score by 50% and CRG by 5.8%, and on external validation, it increases F1-Score by 45.1% and CRG by 5%. These results highlight the importance of structured data and balanced sampling for reliable report generation. Our code is publicly available[1].

**Keywords:** Class imbalance, Structured report generation, Vision-language models.

## 1. Introduction

Radiology reports play a central role in clinical communication, summarizing imaging findings and guiding diagnostic and therapeutic decisions. Automating this process through radiology report generation has emerged as a key challenge in multimodal learning, aiming to bridge visual understanding and clinical language. Recent advances in vision-language models (VLMs) have demonstrated significant progress (Buess et al., 2025b), especially when trained on large datasets pairing computed tomography (CT) volumes with radiologist-written reports (Hamamci et al., 2024; Blankemeier et al., 2026). Moreover, recent studies show that AI-generated draft reports can reduce reporting time by about 25% while maintaining diagnostic accuracy (Acosta et al., 2024), counteracting increasing workload pressures in clinical practice.

Despite recent progress, most existing approaches formulate report generation as free-text prediction (Pellegrini et al., 2025; Hyland et al., 2023), where models directly produce

---

[*] Equal Contribution

1. Code: https://github.com/Kurin-FAU/ALO

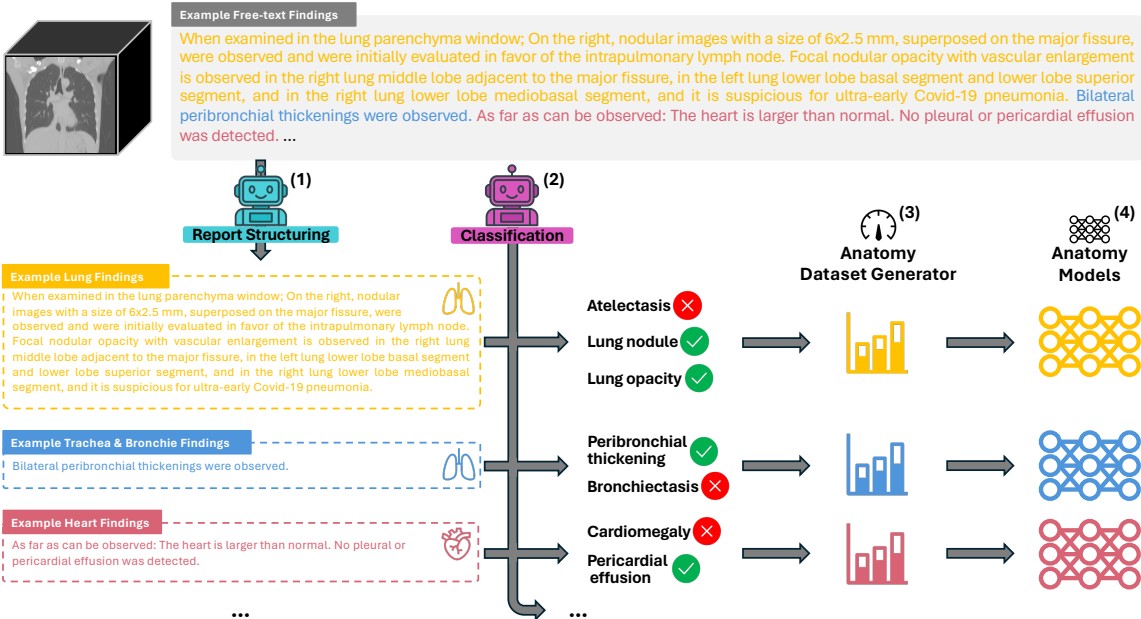

Figure 1: ALO: (1) Convert free-text, patient-level reports into structured anatomy-level findings. (2) Assign normal or abnormal labels to each anatomical region's findings. (3) Construct balanced datasets through targeted oversampling. (4) Train anatomy expert models on the balanced datasets.

narrative reports from images. While intuitive, this setup inherits the variability of clinical writing: syntax, style, and level of detail differ widely across radiologists and institutions, making both learning and evaluation inconsistent. Additionally, radiology datasets are dominated by normal findings (Zhang et al., 2024), creating severe class imbalance that biases models toward underreporting abnormalities, which are the findings most critical for clinical decision-making.

In response, much of the field has focused on architectural or training innovations (Hamamci et al., 2025b; Hein et al., 2025), including large multimodal transformers and increasingly large-scale pretraining (Liu et al., 2025b; Jiang et al., 2025). Nevertheless, supervision quality and label distribution remain persistent challenges that are not resolved by architectural complexity alone. This motivates a complementary perspective: improving the dataset itself through structure and balance can provide substantial gains in diagnostic relevance and reporting consistency, even without modifying model architectures.

Motivated by these observations, we introduce Anatomy-Level Oversampling (ALO) (Figure 1), a simple, yet effective data-centric strategy for structured and balanced report generation. We first organize each free-text report into sections describing individual anatomical regions. To enable balanced sampling, we assign a label to each section indicating the presence of healthy or abnormal findings. ALO then utilizes these labels to apply targeted oversampling, effectively reducing the dominance of normal findings. This setup

provides standardized and balanced supervision across anatomical regions and enables more fine-grained evaluation at the anatomy level instead of the patient level used in most existing works. Because ALO operates entirely at the data level, it is architecture-agnostic and can be easily integrated into existing VLM training pipelines. In addition, the anatomy-level formulation makes the training process modular, allowing individual anatomy models to be retrained or updated without affecting the performance of other anatomies.

We evaluate ALO on three public CT datasets and show that this data-centric strategy substantially improves model sensitivity to abnormal findings and overall reporting performance. Based on these results, our main contributions are:

- We introduce ALO, a simple and model-agnostic strategy that balances healthy and abnormal findings within radiology reports, reducing normal-findings bias and increasing sensitivity to pathologies.

- We present a modular anatomy-level modeling framework in which each anatomical region is trained independently, enabling targeted improvements to individual anatomies without degrading performance on others.

- We perform a comprehensive and fine-grained evaluation of report generation models using anatomy-level assessment and a broad suite of clinical, classification, and natural language generation (NLG) metrics, providing substantially more detailed insights than the patient-level evaluations used in most existing works.

## 2. Related Work

**Radiology Report Generation**: Radiology report generation is a central task in medical multimodal learning, where models aim to translate imaging data into clinically meaningful text. Most existing systems frame the task as free-text report generation, producing full narrative reports directly from images (Pellegrini et al., 2025; Hyland et al., 2023). This direction has been enabled by large paired datasets (Johnson et al., 2019; Zhang et al., 2025) and, more recently, CT-focused resources such as CT-RATE (Hamamci et al., 2026) and Merlin (Blankemeier et al., 2026). Multimodal foundation models and large-scale pre-training (Agrawal et al., 2025; Buess et al., 2025a; Liu et al., 2025b) have further advanced the field by providing stronger visual encoders and more capable language models. Agentic systems (Mao et al., 2025) extend this direction by using large language models (LLMs) to refine or critique reports, improving coherence and clinical correctness.

To address limitations inherent in free-text supervision, several studies have introduced structured formulations. Delbrouck et al. (2025) and Moll et al. (2025) propose the structured radiology report generation task, converting free-text reports into standardized templates to reduce variability and enable clearer evaluation. Keicher et al. (2024) present FlexR, a few-shot classification framework operating on standardized report formats that uses language embeddings for structured prediction with minimal annotation. These efforts demonstrate the value of structured supervision, yet most state-of-the-art systems still rely on free-text, patient-level training without anatomical organization.

**Class Imbalance in Medical AI**: Class imbalance is a well-known challenge in medical image analysis, where normal conditions greatly outnumber abnormal ones. This skew can

bias models toward predicting healthy cases and reduce sensitivity to clinically important pathologies (Salmi et al., 2024). Common mitigation strategies include weighted losses, sampling adjustments, and data augmentation (Chawla et al., 2002; Liu et al., 2025a; Yun et al., 2011), though these are typically used for classification or segmentation tasks. In radiology report generation, class imbalance is harder to address because LLMs are trained using token-wise cross-entropy over patient-level free-text rather than explicit class-level supervision. As a result, standard imbalance-handling techniques such as weighted or focal losses are not directly applicable. Moreover, naïve patient-level oversampling would also amplify normal findings from unrelated anatomical regions, reinforcing the normality bias. To our knowledge, anatomy-level imbalance has received little attention in report generation. Our approach addresses this gap by rebalancing abnormal content at the anatomy level.

## 3. Methods

ALO aims to increase the sensitivity of radiology report generation models to pathological findings by reorganizing the task at the anatomy level and correcting the strong imbalance between normal and abnormal findings. ALO consists of three steps: (1) structuring and labeling reports at the anatomy level, (2) balancing the distribution of normal and abnormal findings, and (3) training anatomy-specific expert models.

### 3.1. Anatomy-Level Structuring and Labeling

To obtain consistent supervision, we convert each free-text, patient-level report into a set of anatomy-level findings (Figure 1) using the anatomy annotations provided by RadGenome-ChestCT (Zhang et al., 2024). Each extracted anatomy-level finding $f_i$ is then labeled as normal or abnormal using a report classifier $C(\cdot)$, which predicts

$$y_i = C(f_i), \qquad y_i \in \{\text{normal}, \text{abnormal}\}.$$

The classifier predicts 18 pathology labels per findings section and assigns an abnormal label if at least one pathology is detected, and normal otherwise.

The complete structured report is written as

$$R = \{(a_i, y_i, f_i)\}_{i=1}^{N},$$

where $a_i$ is the anatomical region, $f_i$ is the extracted findings text for that region, $y_i$ is the predicted label, and $N$ is the total number of anatomical regions considered.

We assess the robustness of the report structuring step via a content preservation analysis on the CT-RATE dataset. Results are reported in Appendix 5.

### 3.2. Balancing Normal and Abnormal Findings

Normal findings are substantially more frequent than abnormal ones. This imbalance encourages models to repeat normal statements while underreporting pathological findings. To reduce this effect, we increase the presence of abnormal samples during training.

For each anatomical region $a$, we construct an anatomy-specific training set $D_a$ consisting of all anatomy-level findings $f_i$ assigned to $a$ together with their corresponding

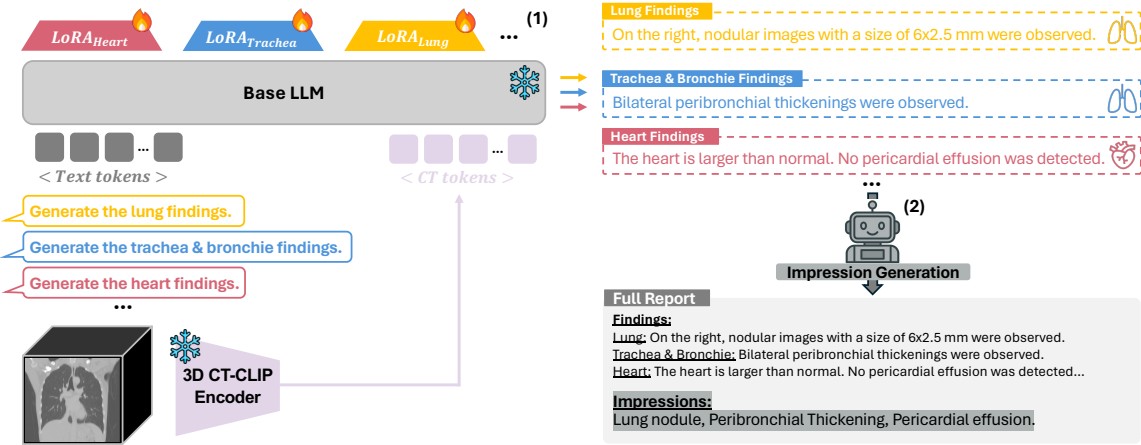

Figure 2: Inference Pipeline: (1) Expert models generate anatomy-level findings. (2) Impression generation model summarizes anatomy-level findings into impressions.

normal/abnormal labels $y_i$. The subset $A_a \subset D_a$ contains all samples with $y_i =$ abnormal. Given an oversampling factor $x \geq 1$, we construct an ALO-balanced dataset

$$D_a^{\text{ALO}} = D_a \cup \underbrace{A_a \cup \cdots \cup A_a}_{x \text{ times}}.$$

In other words, we keep all original samples and add the abnormal subset $A_a$ exactly $x$ additional times, making the ratio between normal and abnormal samples more balanced.

### 3.3. Anatomy-Specific Generation

We reformulate report generation as a modular, anatomy-conditioned prediction task. A 3D visual encoder $E(\cdot)$ processes a CT volume $V$ to produce a visual embedding $v$.

On top of this embedding, we train a set of anatomy-specific expert generators $\{G_i\}_{i=1}^N$, each receiving only the balanced dataset for its corresponding anatomy $a_i$ (see Figure 1). During training, each expert learns to generate the anatomy-level findings section

$$\hat{f}_i = G_i(v),$$

allowing the model to specialize in the visual cues, anatomy-specific report phrasing, and structure characteristic of that anatomy.

At inference time, the volume is encoded once to obtain the visual embedding $v$, which is shared by all anatomy experts. The shared embedding provides each expert with global visual context, supporting systemic disease patterns and cross-anatomical correlations. Each expert then operates individually to produce its anatomy-level findings,

$$\hat{f}_1, \hat{f}_2, \ldots, \hat{f}_N.$$

These findings are concatenated in a fixed anatomical order to form the findings section. A separate language model $I(\cdot)$ acts as an impression agent, converting the findings into a concise clinical summary (see Figure 2):

$$\hat{I} = I\Big(\text{Concat}(\hat{f}_1, \ldots, \hat{f}_N)\Big).$$

The final report mirrors the conventional radiological structure: a detailed anatomy-level findings section followed by a patient-level impression section.

## 4. Experimental Setup

### 4.1. Dataset

**Training.** We train our models on the RadGenome-ChestCT dataset (Zhang et al., 2024), a lightweight structured variant of CT-RATE (Hamamci et al., 2026). RadGenome-ChestCT provides anatomy-level sentences for all CT-RATE reports, enabling the construction of balanced anatomy-specific datasets using ALO (Section 3.2). For all experiments, we set the oversampling factor to 1. Table 1 summarizes the total number of anatomy-level training samples, the prevalence of abnormal findings, and the resulting dataset sizes after applying ALO. Additional pathology distribution statistics are provided in Appendix A.1.

For CT volume preprocessing, we follow CT-CHAT (Hamamci et al., 2026). Volumes are resampled to a uniform voxel spacing of $0.75\,\text{mm} \times 0.75\,\text{mm} \times 1.5\,\text{mm}$ and resized to a fixed shape of $480 \times 480 \times 240$ using center-cropping or padding. HU values are clipped to $[-1000, 1000]$ and normalized to the range $[-1, 1]$.

Table 1: Anatomy-level sample counts in the RadGenome-ChestCT train split (23,880 total samples), including prevalence of abnormal findings, and the ALO dataset size.

| Anatomy | Total Samples | Abnormal | Abnormal % | ALO Dataset |
|---|---|---|---|---|
| Lung | 23,494 | 19,079 | 81.1 | 42,573 |
| Trachea&Bronchi | 21,754 | 1,731 | 7.9 | 23,485 |
| Mediastinum | 23,438 | 9,515 | 40.7 | 32,953 |
| Heart | 23,048 | 6,364 | 27.6 | 29,412 |
| Esophagus | 20,553 | 3,335 | 16.3 | 23,888 |
| Pleura | 17,983 | 6,653 | 37.0 | 24,636 |
| Abdomen | 23,307 | 7,498 | 32.2 | 30,805 |
| Bone | 23,235 | 1,530 | 6.6 | 24,765 |
| Others | 6,210 | 1,343 | 21.6 | 7,553 |

**Validation.** For validation, we use three public datasets covering both internal and external distributions. First, we evaluate on the official CT-RATE validation split (Hamamci et al., 2026), which contains 1,564 studies. We additionally submit predictions to the VLM3D challenge leaderboard (Hamamci et al., 2026), which evaluates performance on a hidden in-center validation set comprising 2,000 patients.

As an external benchmark, we use the RAD-ChestCT dataset (Draelos et al., 2021), which includes 3,630 chest CT studies with 16 pathology labels (label mappings follow

Hamamci et al. (2026) and are reported in Appendix A.2). Because the second external dataset used in CT-RATE (i.e., UPMC) is not publicly accessible, we replace it with AMOS-MM,[2] which provides 510 CT scans covering the chest (Ji et al., 2022). To ensure compatibility with CT-RATE, we extract only chest slices and reports. The pathology labels are obtained using the CT-RATE report classifier[3]. More details about AMOS-MM preprocessing can be found in Appendix A.3.

## 4.2. Baseline Methods

We compare ALO against four baselines that share the same model architecture, vision encoder, and training setup. (1) CT-CHAT is a public model trained on 2.7 million question-answer pairs which also include free-text reports from CT-RATE. (2) Free-Text trains the model on free-text, patient-level reports. (3) Structured uses anatomy-level report decomposition but trains a single model on all anatomies without balancing. (4) Anatomy Experts trains separate anatomy expert models on the anatomy-specific findings sections while preserving the original class imbalance.

## 4.3. Evaluation

We evaluate all models using the VLM3D challenge protocol[4] for classification-based metrics and complement it with RadEval (Xu et al., 2025), to provide a comprehensive set of clinical and NLG measures. Performance is assessed at both patient and anatomy levels. Following the VLM3D protocol, we evaluate both the findings and impressions sections. For CT-RATE and AMOS-MM, we report both RadEval and VLM3D metrics. For RAD-ChestCT, we report only classification-based metrics because textual reports are not available. In our analysis, we highlight Precision, Recall, F1-score (Table 2), and pathology-level metrics (Figures 3 and 4), as these better reflect sensitivity to abnormal findings in imbalanced report generation settings, while still reporting the full set of clinical, NLG, and classification metrics for completeness.

## 4.4. Implementation Details

**Findings VLM.** We finetune Meta-Llama-3.1-8B-Instruct[5] with CT-CLIP[6] as a frozen vision encoder. Anatomy expert models are trained independently with LoRA adapters (Hu et al., 2022). Training follows Adam with a cosine schedule (lr $2 \times 10^{-5}$), effective batch size 16, for 10 epochs on four NVIDIA A100 (80 GB) GPUs.

**Impressions LLM.** For generating the patient-level impressions section from the anatomy-level findings, we finetune SmolLM3-3B[7] using Axolotl (Axolotl maintainers and contributors, 2023) for 3 epochs with Adam and a cosine schedule (lr $1 \times 10^{-4}$), using an effective batch size of 1,024 on four NVIDIA A40 (40 GB) GPUs.

---

2. AMOS-MM Dataset: https://era-ai-biomed.github.io/amos/dataset.html#overview

3. Report classifier: https://huggingface.co/datasets/ibrahimhamamci/CT-RATE/tree/main/models

4. VLM3D challenge: https://reportgen.vlm3dchallenge.com

5. VLM findings LLM: https://huggingface.co/meta-llama/Llama-3.1-8B-Instruct

6. CT-CLIP: https://huggingface.co/datasets/ibrahimhamamci/CT-RATE/tree/main/models/

7. Impressions LLM: https://huggingface.co/HuggingFaceTB/SmolLM3-3B

## 5. Results and Discussion

We first assess the impact of ALO on patient-level across three datasets: CT-RATE (internal validation), RAD-ChestCT (external validation), and AMOS-MM (external validation). The main clinical, NLG, and classification metrics are summarized in Table 2.

Table 2: Patient-level performance of baseline models and ALO-enhanced variant across three evaluation datasets, reported using clinical, NLG (Natural Language Generation), and classification (CL) metrics. Classification metrics are reported with 95% confidence intervals. (**bold** = best on dataset; ▨ highlighted columns show VLM3D relevant metrics).

| Dataset | Method | Clinical ↑ | | | | | NLG ↑ | | CL (macro) ↑ | | |
|---|---|---|---|---|---|---|---|---|---|---|---|
| | | GREEN | RaTE | RadGraph | 1/RadCLIQ | CRG | BLEU | BERT | P | R | F1 |
| **CT-RATE** (1,564 scans) | CT-CHAT | 0.437 | 0.664 | 0.200 | 1.235 | 0.367 | 0.203 | 0.611 | 0.354 [0.307, 0.400] | 0.158 [0.150, 0.165] | 0.169 [0.161, 0.176] |
| | Free-Text | 0.435 | 0.659 | 0.201 | 1.225 | 0.353 | 0.201 | 0.612 | **0.389** [0.331, 0.425] | 0.097 [0.091, 0.104] | 0.115 [0.106, 0.124] |
| | Structured | **0.489** | **0.678** | **0.232** | **1.276** | 0.356 | 0.218 | 0.615 | 0.356 [0.332, 0.383] | 0.118 [0.109, 0.127] | 0.168 [0.156, 0.180] |
| | Anatomy Experts | 0.480 | 0.675 | 0.216 | 1.246 | 0.364 | 0.208 | **0.617** | 0.349 [0.328, 0.371] | 0.147 [0.139, 0.156] | 0.190 [0.180, 0.199] |
| | ALO | 0.341 | 0.662 | 0.197 | 1.171 | **0.385** | **0.219** | 0.604 | 0.332 [0.318, 0.346] | **0.260** [0.250, 0.270] | **0.285** [0.274, 0.295] |
| **RAD-ChestCT** (3,630 scans) | CT-CHAT | - | - | - | - | **0.385** | - | - | 0.320 [0.296, 0.342] | 0.178 [0.173, 0.183] | 0.173 [0.167, 0.179] |
| | Free-Text | - | - | - | - | 0.362 | - | - | 0.352 [0.314, 0.396] | 0.114 [0.109, 0.119] | 0.130 [0.124, 0.136] |
| | Structured | - | - | - | - | 0.348 | - | - | 0.355 [0.329, 0.382] | 0.081 [0.075, 0.087] | 0.122 [0.114, 0.130] |
| | Anatomy Experts | - | - | - | - | 0.363 | - | - | **0.409** [0.366, 0.451] | 0.133 [0.126, 0.140] | 0.175 [0.166, 0.182] |
| | ALO | - | - | - | - | 0.381 | - | - | 0.328 [0.317, 0.340] | **0.227** [0.218, 0.235] | **0.254** [0.246, 0.262] |
| **AMOS-MM** (510 scans) | CT-CHAT | 0.197 | **0.513** | 0.035 | **0.635** | 0.339 | 0.025 | **0.432** | **0.182** [0.092, 0.215] | 0.142 [0.115, 0.168] | 0.086 [0.070, 0.103] |
| | Free-Text | 0.215 | 0.506 | 0.033 | 0.627 | 0.341 | 0.022 | 0.425 | 0.179 [0.072, 0.235] | 0.048 [0.035, 0.058] | 0.044 [0.032, 0.058] |
| | Structured | 0.209 | 0.507 | 0.032 | 0.606 | 0.349 | 0.019 | 0.399 | 0.147 [0.119, 0.176] | 0.118 [0.080, 0.134] | 0.110 [0.086, 0.134] |
| | Anatomy Experts | **0.230** | 0.510 | 0.036 | 0.621 | 0.353 | 0.022 | 0.419 | 0.157 [0.128, 0.186] | 0.118 [0.089, 0.156] | 0.103 [0.083, 0.121] |
| | ALO | 0.174 | **0.513** | **0.039** | 0.623 | **0.379** | **0.027** | 0.420 | 0.176 [0.153, 0.200] | **0.214** [0.184, 0.248] | **0.166** [0.146, 0.185] |

On the internal CT-RATE split, ALO substantially improves sensitivity to pathological findings. Compared to the Anatomy Experts baseline without oversampling, Recall increases from 0.147 to 0.260 and F1-Score from 0.190 to 0.285, while Precision only decreases slightly. Models trained on patient-level free-text or patient-level structured reports yield higher traditional clinical metrics (e.g., GREEN, RaTE, RadGraph), with the structured model achieving the strongest overall clinical scores. At the same time, the ALO-enhanced model attains the highest CRG score (0.385), a distribution-aware metric that emphasizes clinically relevant abnormalities and mitigates the tendency of conventional clinical metrics (e.g., GREEN) to favor trivial or normal-dominated predictions (Hamamci et al., 2025a). Classic NLG metrics (BLEU and BERTScore) remain stable across all variants, indicating that ALO primarily affects clinical correctness rather than surface-level fluency or style.

The gains in Recall and F1-Score generalize to external datasets. On RAD-ChestCT, Recall improves from 0.178 (CT-CHAT) and 0.133 (Anatomy Experts) to 0.227 with ALO, with a corresponding F1-Score increase from 0.173 (CT-CHAT) and 0.175 (Anatomy Experts) to 0.254. On AMOS-MM, ALO again achieves the strongest clinical performance, outperforming free-text and structured baselines with higher CRG (0.379), as well as the best Recall (0.214) and F1-Score (0.166).

Taken together, these results demonstrate that balancing anatomy-level supervision via ALO systematically improves abnormality detection on both internal and external datasets, while preserving the overall text quality and structure of the generated reports. Additional results, including our VLM3D challenge submission and extended anatomy-level analyses, are provided in Appendices B.2 and B.3.

### 5.1. Ablation Study: Effect of Anatomy-Level Oversampling

To isolate the contribution of ALO, we compare the Anatomy Experts baseline to its ALO-enhanced variant on the internal CT-RATE validation set and the external RAD-ChestCT dataset (AMOS-MM results can be found in Appendix B.1). Figures 3 and 4 show radar plots of per-pathology Precision, Recall, and F1-Score for both training strategies. Each axis corresponds to a specific pathology and the curves represent the Anatomy Experts baseline and the ALO-trained model.

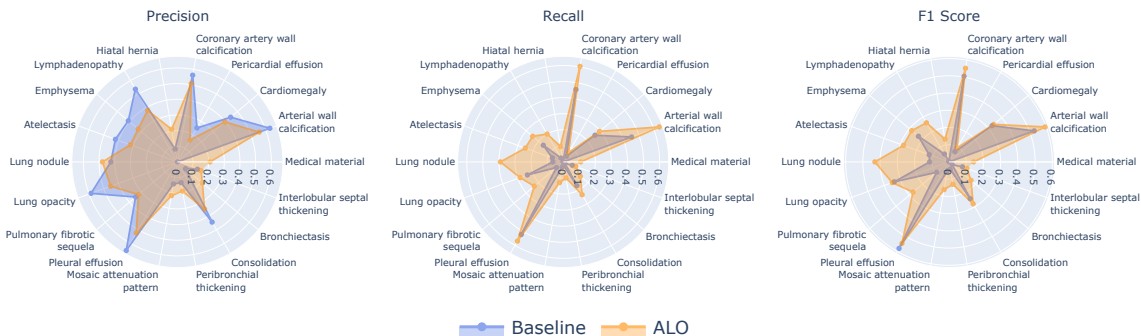

Figure 3: Per-pathology ablation on the internal CT-RATE dataset. The radar plots show Precision, Recall, and F1-Score for each pathology, comparing the Anatomy Experts baseline with the ALO-trained model.

On the internal CT-RATE split, the ALO curve consistently encloses the Anatomy Experts baseline for Recall and F1-Score across nearly all pathologies, indicating a systematic reduction in false negatives. At the same time, Precision drops only slightly, suggesting that oversampling introduces minimal additional false positives (see Figure 3). This effect stems from anatomy-level oversampling, which favors abnormality detection, slightly increasing false positives while substantially reducing false negatives. For time-critical findings such as pulmonary nodules, higher recall is clinically preferable.

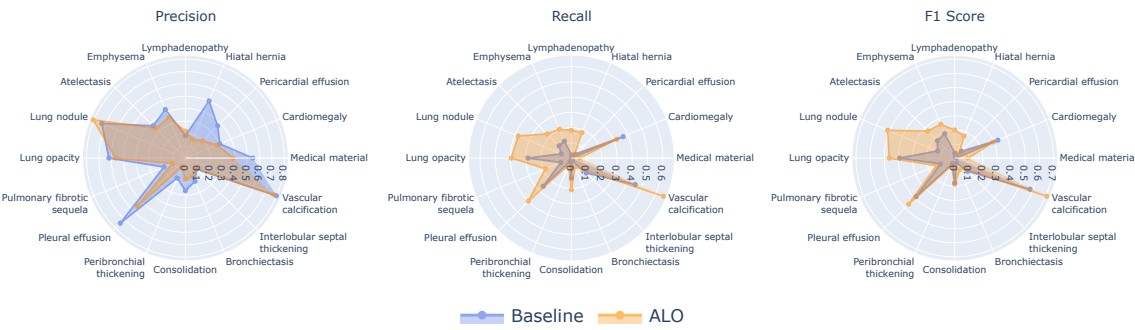

Figure 4: Per-pathology ablation on the external RAD-ChestCT dataset. The radar plots show Precision, Recall, and F1-Score for each pathology, comparing the Anatomy Experts baseline with the ALO-trained model.

The external RAD-ChestCT ablation shows a similar pattern. ALO improves Recall and F1-Score for several clinically important conditions. Again, Precision remains largely stable compared to the Anatomy Experts baseline. This consistency across datasets supports the conclusion that ALO primarily improves model sensitivity.

## 5.2. Qualitative Evaluation

Figure 5 shows outputs from the lung and heart expert models. The generated reports successfully identify major pathologies, such as "nonspecific nodule" and "increased heart size" (True Positives). However, discrepancies remain: for instance, the heart model misses the "atherosclerotic wall calcifications" (False Negative), consistent with the low overall F1-scores in Table 2 and highlighting that current report generation models still struggle with comprehensive abnormality detection. Furthermore, differences in reported normal findings (Figure 5 "Non-Overlapping Normal Findings") contribute to lower clinical and NLG scores despite correct identification of primary findings, underscoring the limitation of existing metrics in capturing clinical correctness.

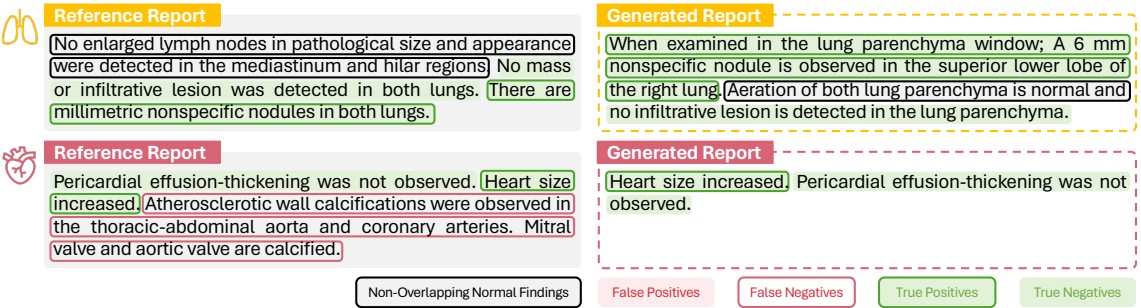

Figure 5: Case study for lung and heart. Green marks correct detections (true positive/negative); red marks diagnostic errors (false positive/negative); "Non-Overlapping Normal Findings" shows valid healthy findings differing from the reference.

## 5.3. Limitations

While ALO yields substantial gains, we observe three limitations. First, the modular architecture increases inference complexity by requiring separate forward passes for each anatomy and the impressions model, resulting in an $1.6\times$ increase in inference time compared to single-pass baselines. However, this design improves overall clinical performance and enables independent updates for specific anatomy expert models. Second, the pipeline relies on upstream tools for report structuring and labeling. While errors introduced at this stage may propagate to downstream training, quantifying the impact of structuring noise remains an open question for future work. Finally, the external AMOS-MM dataset presents a significant domain gap: as an abdomen-focused dataset, its chest reports are substantially shorter and stylistically distinct from the thoracic-focused CT-RATE, presenting a challenging out-of-distribution test.

## 6. Conclusion

In this work, we showed that radiology report generation models continue to struggle with two fundamental issues: the variability of free-text supervision and the strong imbalance between normal and abnormal findings. These limitations reduce the consistency of supervision and bias models toward underreporting clinically important pathologies. We addressed these issues through ALO, a simple data-centric strategy that restructures reports into anatomy-level findings sections and balances the data distribution within each region. Combined with modular anatomy expert models, this approach provides stable supervision and effectively mitigates the dominance of normal findings. Experiments across three public datasets show that ALO substantially improves sensitivity and F1-Score. Our results highlight that a data-centric view of report generation can meaningfully improve clinical reliability. Structuring reports at the anatomy level not only enables targeted oversampling but also allows anatomy-level evaluation, offering more precise insights into model strengths and weaknesses. Ultimately, our work demonstrates that for report generation, improving the supervision signal can be as impactful as architectural innovations.

## Acknowledgments

The authors gratefully acknowledge the scientific support and HPC resources provided by the Erlangen National High Performance Computing Center (NHR@FAU) of the Friedrich-Alexander-Universität Erlangen-Nürnberg (FAU). The hardware is funded by the German Research Foundation (DFG).

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

## Contents

## Appendix A.  Data

### A.1.  CT-RATE / RadGenome-ChestCT Label Distribution

Table 3 summarizes the prevalence of the 18 pathology labels in the RadGenome-ChestCT training and validation splits. The distribution is clearly imbalanced, with some pathologies occurring far more frequently than others and a skewed ratio between healthy and pathological cases across labels.

Table 3: Pathology counts in the 23,880 training and 1,552 validation reports.

| Pathology | Train | Val | Train Ratio | Val Ratio |
|---|---|---|---|---|
| Medical material | 2,811 | 151 | 0.118 | 0.097 |
| Arterial wall calcification | 6,570 | 420 | 0.275 | 0.271 |
| Cardiomegaly | 2,480 | 156 | 0.104 | 0.101 |
| Pericardial effusion | 1,654 | 104 | 0.069 | 0.067 |
| Coronary artery calcification | 5,856 | 378 | 0.245 | 0.244 |
| Hiatal hernia | 3,386 | 215 | 0.142 | 0.139 |
| Lymphadenopathy | 6,023 | 389 | 0.252 | 0.251 |
| Emphysema | 4,633 | 300 | 0.194 | 0.193 |
| Atelectasis | 6,076 | 356 | 0.254 | 0.229 |
| Lung nodule | 10,874 | 680 | 0.455 | 0.438 |
| Lung opacity | 8,788 | 598 | 0.368 | 0.385 |
| Pulmonary fibrotic sequela | 6,368 | 410 | 0.267 | 0.264 |
| Pleural effusion | 2,818 | 179 | 0.118 | 0.115 |
| Mosaic attenuation pattern | 1,748 | 124 | 0.073 | 0.080 |
| Peribronchial thickening | 2,454 | 164 | 0.103 | 0.106 |
| Consolidation | 4,203 | 286 | 0.176 | 0.184 |
| Bronchiectasis | 2,402 | 161 | 0.101 | 0.104 |
| Interlobular septal thickening | 1,868 | 121 | 0.078 | 0.078 |

## A.2. Rad-ChestCT Label Mapping

To ensure consistent evaluation across datasets, we align the CT-RATE pathology labels with the more fine-grained annotation schema of RAD-ChestCT. Table 4 shows the mapping used in our experiments. Several CT-RATE labels correspond to multiple RAD-ChestCT labels (e.g., Medical material, Lung nodule), which we merge into a single binary label to maintain compatibility with the CT-RATE taxonomy. This harmonization enables the use of the identical classifier and evaluation metrics for both datasets.

Table 4: Label mapping between CT-RATE and RAD-ChestCT datasets.

| CT-RATE Label | RAD-ChestCT Label |
|---|---|
| Medical material | pacemaker_or_defib, catheter_or_port, hardware, stent, suture, staple, chest_tube, tracheal_tube, gi_tube, breast_implant, heart_valve_replacement, clip |
| Arterial wall calcification | calcification, scattered_calc |
| Cardiomegaly | cardiomegaly |
| Pericardial effusion | pericardial_effusion |
| Coronary artery wall calcification | calcification, scattered_calc |
| Hiatal hernia | hernia |
| Lymphadenopathy | lymphadenopathy |
| Emphysema | emphysema |
| Atelectasis | atelectasis |
| Lung nodule | nodule, nodulegr1cm, scattered_nod |
| Lung opacity | opacity |
| Pulmonary fibrotic sequela | fibrosis |
| Pleural effusion | pleural_effusion |
| Mosaic attenuation pattern | all_zeros |
| Peribronchial thickening | bronchial_wall_thickening |
| Consolidation | consolidation |
| Bronchiectasis | bronchiectasis |
| Interlobular septal thickening | septal_thickening |

## A.3. AMOS-MM Processing

**Volume Processing.** AMOS-MM contains thoracoabdominal CT volumes with reports covering chest, abdomen, and pelvis. To align the dataset with the chest-focused CT-RATE setup, we extract only the thoracic region using the following steps:

1. Retain studies that include a chest-related report section.

2. Run TotalSegmentator (Wasserthal et al., 2023) to isolate thoracic region defined by:
   `lung_upper_lobe_left`, `lung_lower_lobe_left`, `lung_upper_lobe_right`,
   `lung_middle_lobe_right`, `lung_lower_lobe_right`, `esophagus`, `trachea`

3. Crop the volume to the thoracic bounding box, dropping non-thoracic slices.

**Report Processing.** For anatomy-level evaluation, we convert the free-text AMOS-MM chest reports into the structured anatomy-level format (following RadGenome-ChestCT). We use the GPT-4.1 (2025-04-14) model via Azure OpenAI Services. The model extracts each sentence from the report and assigns predefined anatomies to the sentence (Figure 6).

```
AMOS-MM Structuring Prompt

You are a radiologist tasked with extracting anatomical regions from the findings
section of radiology reports. For each sentence provided, identify the corresponding
anatomical regions. Ensure each identified region is an entry from a predefined list:
[", ".join(ANATOMY_LIST)]

If a sentence mentions 'left' or 'right', these qualifiers should precede the
anatomical region (e.g., left kidney). Given input in the format:
<Input><findings><\Input>.

Please reply in the following JSON format:
{<sentence>: [region1,region2,...], <sentence>: [region1]}.

Findings: {findings}
```

Figure 6: GPT-4.1 report structuring prompt.

**Findings:** "A few speckled slightly high-density lesions can be seen in the right upper lobe of the lung and the left lower lobe, with unclear boundaries. Local transparency is increased in the right lung. The trachea and bronchi are unobstructed. The size and shape of the heart and great blood vessels are normal. Local pleural thickening on both sides."

```
Example GPT-4.1 Response

"result": {
    "A few speckled slightly high-density lesions can be seen in the right upper lobe
    of the lung and the left lower lobe, with unclear boundaries.": [
        "right lung",
        "left lung"
    ],
    "Local transparency is increased in the right lung.": [
        "right lung"
    ],
    "The trachea and bronchi are unobstructed.": [
        "trachea and bronchi"
    ],
    "The size and shape of the heart and great blood vessels are normal.": [
        "heart"
    ],
    "Local pleural thickening on both sides.": [
        "pleura"
    ],
}
```

Figure 7: GPT-4.1 report structuring example response.

### A.4. Report Structuring Evaluation

To assess the robustness of the report structuring stage, we evaluate content preservation on CT-RATE using 1,000 samples from the training split. Structured reports derived from ground-truth free-text reports are compared against the original reports using the same clinical, NLG, and classification metrics as in the main evaluation, verifying that the structuring process preserves the underlying content.

Table 5: Report structuring evaluation on 1,000 CT-RATE training samples. High scores indicate that report content is preserved after structuring.

| Clinical ↑ | | | | | NLG ↑ | | CL (macro) ↑ | | |
|---|---|---|---|---|---|---|---|---|---|
| GREEN | RaTE | RadGraph | 1/RadCLIQ | CRG | BLEU | BERT | P | R | F1 |
| 0.959 | 0.997 | 0.706 | - | 0.907 | 0.877 | 0.885 | 0.978 | 0.951 | 0.964 |

## Appendix B. Extended Results

### B.1. Ablation Study

Figure 8 shows the ablation results for the Anatomy Experts baseline and ALO-enhanced model on AMOS-MM. ALO yields consistent improvements in recall-oriented clinical metrics while maintaining comparable precision.

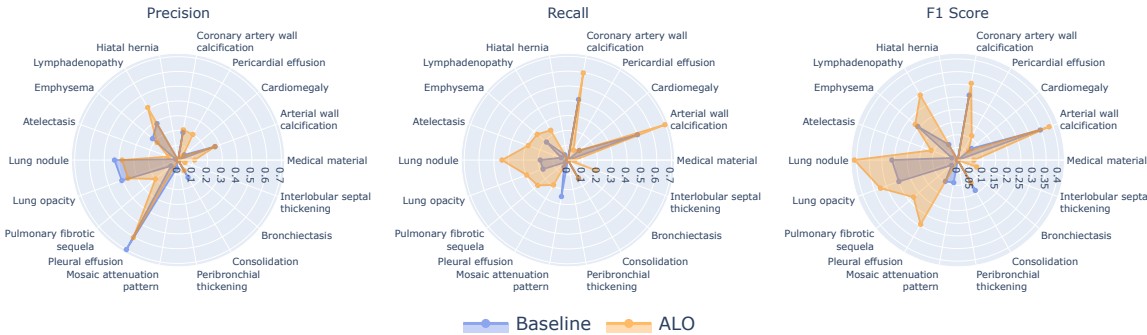

Figure 8: Per-pathology ablation on the external AMOS-MM dataset. The radar plots show Precision, Recall, and F1-Score for each pathology, comparing the Anatomy Experts baseline with the ALO-trained model.

## B.2. VLM3D Challenge Results

Table 6 summarizes the results on the hidden VLM3D challenge test set. We report the performance of the three submitted systems: CT-CHAT, our Anatomy Experts baseline, and our ALO-based model (an earlier version of the method presented in this paper). The table includes all clinical, NLG, and classification metrics used by the challenge for ranking.

Table 6: Results on the hidden VLM3D challenge test set.

| Dataset | Method | Clinical Metrics ↑ | NLG Metrics ↑ | | CL (macro) ↑ | | |
|---|---|---|---|---|---|---|---|
| | | CRG | BLEU | ROUGE | P | R | F1 |
| **CT-RATE** (2,000 scans) | CT-CHAT | 0.380 | **0.265** | **0.249** | 0.233 | **0.329** | 0.176 |
| | Anatomy Experts | 0.366 | 0.240 | 0.230 | **0.380** | 0.157 | 0.201 |
| | ALO | **0.383** | 0.259 | 0.232 | 0.342 | 0.260 | **0.288** |

## B.3. Anatomy-Level Evaluation

Tables 7 and 8 present per-anatomy metrics for the Anatomy Experts baseline and the ALO-enhanced model on the CT-RATE and AMOS-MM datasets.

Table 7: Per-anatomy metrics for Anatomy Experts baseline (without ALO).

| Dataset | Anatomy | Clinical Metrics ↑ | | | | NLG Metrics ↑ | |
|---|---|---|---|---|---|---|---|
| | | GREEN | RaTE | RadGraph | 1/RadCLIQ | BLEU | BERT |
| **CT-RATE** (1,564 scans) | Lung | 0.261 | 0.580 | 0.127 | 1.114 | 0.128 | 0.575 |
| | Trachea&Bronchi | 0.777 | 0.833 | 0.578 | - | 0.212 | 0.740 |
| | Mediastinum | 0.615 | 0.700 | 0.152 | 1.796 | 0.200 | 0.598 |
| | Heart | 0.569 | 0.615 | 0.378 | 2.420 | 0.210 | 0.603 |
| | Esophagus | 0.873 | 0.882 | 0.437 | - | 0.525 | 0.837 |
| | Pleura | 0.446 | 0.613 | 0.356 | 1.612 | 0.061 | 0.686 |
| | Bone | 0.364 | 0.743 | 0.345 | 2.903 | 0.133 | 0.580 |
| | Abdomen | 0.372 | 0.586 | 0.156 | 1.418 | 0.235 | 0.571 |
| | Others | 0.085 | 0.345 | 0.070 | 0.843 | 0.033 | 0.367 |
| **AMOS-MM** (510 scans) | Lung | 0.029 | 0.449 | 0.024 | 0.625 | 0.009 | 0.373 |
| | Trachea&Bronchi | 0.143 | 0.523 | 0.011 | 0.523 | 0.000 | 0.363 |
| | Mediastinum | 0.153 | 0.512 | 0.042 | 0.684 | 0.010 | 0.431 |
| | Heart | 0.108 | 0.394 | 0.055 | 0.573 | 0.000 | 0.310 |
| | Esophagus | 0.035 | 0.424 | 0.000 | 0.589 | 0.000 | 0.364 |
| | Pleura | 0.075 | 0.387 | 0.027 | 0.535 | 0.000 | 0.337 |
| | Bone | 0.004 | 0.299 | 0.003 | 0.517 | 0.000 | 0.260 |
| | Abdomen | 0.019 | 0.375 | 0.019 | 0.635 | 0.012 | 0.373 |
| | Others | 0.001 | 0.413 | 0.000 | 0.477 | 0.000 | 0.195 |

Table 8: Per-anatomy metrics applying ALO.

| Dataset | Anatomy | Clinical Metrics ↑ | | | | NLG Metrics ↑ | |
|---|---|---|---|---|---|---|---|
| | | GREEN | RaTE | RadGraph | 1/RadCLIQ | BLEU | BERT |
| **CT-RATE** (1,564 scans) | Lung | 0.203 | 0.552 | 0.090 | 1.069 | 0.134 | 0.553 |
| | Trachea&Bronchi | 0.773 | 0.835 | 0.560 | - | 0.300 | 0.739 |
| | Mediastinum | 0.570 | 0.686 | 0.137 | 1.624 | 0.178 | 0.588 |
| | Heart | 0.547 | 0.607 | 0.350 | 2.063 | 0.192 | 0.597 |
| | Esophagus | 0.844 | 0.870 | 0.409 | - | 0.493 | 0.821 |
| | Pleura | 0.372 | 0.582 | 0.301 | 1.299 | 0.077 | 0.077 |
| | Bone | 0.359 | 0.736 | 0.342 | 2.846 | 0.138 | 0.580 |
| | Abdomen | 0.387 | 0.580 | 0.139 | 1.457 | 0.236 | 0.574 |
| | Others | 0.076 | 0.337 | 0.062 | 0.817 | 0.032 | 0.357 |
| **AMOS-MM** (510 scans) | Lung | 0.019 | 0.421 | 0.019 | 0.642 | 0.202 | 0.391 |
| | Trachea&Bronchi | 0.144 | 0.447 | 0.014 | 0.534 | 0.000 | 0.372 |
| | Mediastinum | 0.109 | 0.521 | 0.032 | 0.702 | 0.011 | 0.434 |
| | Heart | 0.082 | 0.399 | 0.044 | 0.591 | 0.000 | 0.336 |
| | Esophagus | 0.045 | 0.424 | 0.004 | 0.596 | 0.000 | 0.367 |
| | Pleura | 0.060 | 0.402 | 0.027 | 0.572 | 0.000 | 0.361 |
| | Bone | 0.002 | 0.299 | 0.003 | 0.525 | 0.000 | 0.269 |
| | Abdomen | 0.003 | 0.366 | 0.015 | 0.666 | 0.016 | 0.666 |
| | Others | 0.000 | 0.396 | 0.000 | 0.487 | 0.000 | 0.216 |

