# OpenReview forum: "ALO: Addressing Class Imbalance in Radiology Report Generation through Anatomy-Level Oversampling"
_MIDL.io/2026/Conference — MIDL 2026 Poster_

### Official Review · Reviewer_DQGB · 2026-01-06

**Confidence:** 3
**Preliminary Rating:** 4
**Final Rating:** 4

**Summary:**

This paper addresses a critical challenge in the domain of radiology report parsing: the significant class imbalance between abundant normal anatomical findings and scarce abnormal findings. To mitigate this, the authors propose a novel method that identifies anatomical-based features to guide an oversampling strategy. They demonstrate that this approach leads to significant improvements in both F1-score and sensitivity.

**Strengths:**

The paper is well-motivated and the related work section provides excellent context for the study. The efficacy of the ALO method is supported by thorough evaluation and ablation studies, specifically showing marked improvements in F1 performance. Additionally, the authors successfully demonstrate the generalizability of their approach by testing it across various models and datasets.

**Weaknesses:**

* **Methodological Detail:** Certain sections of the methodology require more granular explanation to improve technical transparency. Specifically, the process of mapping anatomically classified texts to the oversampling logic should be described in greater detail to allow for better reproducibility.

* **Contextual Comparison with Existing Imbalance Techniques:** While the authors briefly mention existing methods for addressing class imbalance in medical AI, a deeper discussion or direct empirical comparison is missing. The paper would be strengthened by situating the proposed method against standard techniques. This would clarify whether the anatomical-based approach offers unique advantages over these established baseline strategies.

* **Analysis of Precision Trade-offs:** The ablation studies indicate that while the F1-score and sensitivity improve with ALO, there is a noticeable decrease in Precision. The manuscript currently lacks a thorough discussion of why this trade-off occurs. Expanding on this—perhaps by analyzing whether the oversampling introduces false positives is crucial for a balanced understanding of the model's performance in a clinical context.

**Detailed Comments:**

**Minor improvements**

1. The captions or text descriptions for Figure 1 and Figure 2 refer to specific numbered steps; however, these are not explicitly labeled within the diagrams themselves.

2. There are several instances throughout the manuscript where opening quotation marks are inverted (appearing as closing marks). These should be corrected (e.g., using `` and '' in LaTeX).

**Justification Of Final Rating:**

Thank you for addressing my further feedback. Since the paper was already given a 'weak accept' during the preliminary round and was in good form then, I will be sticking with that rating. All the best!

**Justification Of The Preliminary Rating:**

This paper presents a well-motivated solution to the class imbalance problem in radiology report parsing through an anatomically informed oversampling method. Its primary strength is the demonstrated generalizability and F1-score improvement across multiple architectures and datasets. However,  limited methodological detail regarding the anatomical mapping logic and a lack of discussion on other class imbalance mitigation techniques need to be addressed. Furthermore, the observed decrease in precision during ablation studies requires a more rigorous discussion of the trade-off between sensitivity and false positives to ensure clinical utility.

**Questions To Address In The Rebuttal:**

Please refer to the weaknesses

---

> ### Author Response · Authors · 2026-01-25
> **Response to Reviewer DQGB**
>
> We thank Reviewer DQGB for the positive assessment of our work, in particular for highlighting the strong motivation, the thorough evaluation and ablation studies, and the demonstrated generalizability across datasets. We address the reviewer’s comments below.
>
>
> ### [1] *“Methodological Detail”*
> > We agree that additional methodological detail improves technical transparency, particularly regarding the anatomy-level oversampling logic. We have expanded the description of how anatomically structured findings are mapped to the oversampling procedure in the Methods section, clarifying the construction of anatomy-specific datasets and the application of oversampling at the region level. The added clarifications are highlighted in red in the revised manuscript (Section 3).
>
>
> ### [2] *“Relation to and distinction from standard class-imbalance techniques”*
> > We agree that clearer positioning with respect to existing imbalance-handling techniques strengthens the paper. Standard methods such as weighted loss or focal loss are designed for end-to-end classification with explicit class-level supervision and are therefore not directly applicable to LLM-based free-text report generation, which is trained using token-wise cross-entropy over long textual outputs, with pathology labels used only for post-hoc evaluation.
>
> > Moreover, class balancing in patient-level free-text report generation is not straightforward: naïve patient-level oversampling would also increase normal findings from unrelated anatomical regions, thereby reinforcing the normality bias. ALO addresses this limitation by rebalancing abnormal content per anatomy at the data and supervision level, enabling targeted imbalance correction without amplifying normal findings elsewhere. To our knowledge, such anatomy-level balancing has not been explored in prior work on radiology report generation. We have clarified this distinction in Section 2 (Related Work) of the revised manuscript.
>
>
> ### [3] *“Analysis of precision–recall trade-offs introduced by ALO”*
> > We agree that ALO introduces a precision–recall trade-off. By increasing the relative presence of abnormal findings during training, ALO reduces the strong normality bias in report generation, leading to a slight decrease in precision but a substantially larger gain in recall. As a result, the overall F1-score improves, reflecting a favorable balance between false positives and false negatives. From a clinical perspective, high sensitivity to time-critical pathologies such as pulmonary nodules or pneumothorax is often more important. This precision–recall trade-off and the resulting recall gains are consistently observed across multiple clinically relevant pathologies in Figures 3 and 4.
>
>
> ### *“Minor Improvements”*
> > We thank the reviewer for these helpful suggestions, which improve the clarity and presentation of the paper. We labeled the steps in Figures 1 and 2 and corrected the quotation mark formatting throughout the manuscript.

---

### Official Review · Reviewer_Pt1S · 2026-01-09

**Confidence:** 4
**Preliminary Rating:** 4

**Summary:**

The authors address the issue of class imbalance in radiology report generation, where the dominance of normal findings leads models to underreport clinically significant abnormalities. They propose Anatomy-Level Oversampling (ALO), a data-centric pipeline that first restructures free-text reports into anatomy-specific sections and then balances the training data by oversampling abnormal findings for each anatomical region. This balanced data is used to train modular, anatomy-specific expert models (using LoRA adapters on a Llama-3 base with a CT-CLIP encoder), followed by an impression generation module. The method is evaluated on three datasets: the internal CT-RATE validation set and two external datasets (RAD-ChestCT and AMOS-MM). The results demonstrate that ALO significantly improves sensitivity (Recall) and F1-scores for pathological findings compared to patient-level and non-balanced baselines.

**Strengths:**

Effective Data-Centric Solution: The paper targets a problem in medical AI—class imbalance—with a straightforward, model-agnostic solution. By shifting the focus from architectural complexity to supervision quality (structuring and balancing), the authors demonstrate substantial performance gains (e.g., increasing F1-Score by ≈50% on internal validation). This aligns well with the practical needs of clinical deployment.

Comprehensive Evaluation: The evaluation is rigorous, employing a wide range of metrics including standard NLG metrics (BLEU, BERTScore), classification metrics (Precision, Recall, F1), and specialized clinical metrics (Green, RadGraph, CRG). The inclusion of both internal (CT-RATE) and external (RAD-ChestCT, AMOS-MM) validation sets significantly strengthens the claims regarding generalization.

Clear Presentation: The paper is well-structured and clearly written. The motivation is evident, the methodology is explained with sufficient detail (including the prompt designs in the appendix), and the analysis of results (including the radar plots in Figures 3 and 4) provides good insight into the trade-offs between precision and recall.

**Weaknesses:**

Pipeline Complexity and Error Propagation: The proposed method relies heavily on upstream components for report structuring and labeling. Specifically, it depends on the quality of the "Anatomy Dataset Generator". Any errors in this initial structuring or classification step will propagate to the expert models. While the authors acknowledge this limitation, a quantitative analysis of the structuring error rate is missing.

Loss of Inter-Anatomical Context: By training anatomy experts completely independently, the model potentially loses context regarding systemic pathologies that affect multiple organs simultaneously (e.g., heart failure leading to pleural effusion and lung opacities). While the "Impression Generation" module aims to synthesize these, the initial feature extraction and finding generation are isolated.

Inference Overhead: The architecture requires running multiple LoRA adapters (one per anatomy) and a final impression model. While LoRA is parameter-efficient, the inference pipeline appears to require sequential or parallel execution of multiple generation passes, which could increase latency compared to a single end-to-end patient-level model. The paper does not discuss the computational cost or latency at inference time.

Technical Novelty: The core technical contribution—oversampling the minority class—is a standard machine learning technique. The novelty lies primarily in its application context (anatomy-level structuring for report generation).

**Detailed Comments:**

It would be interesting to know if the ALO model tends to generate fewer normal findings overall because it was trained on a distribution where normal findings were artificially down-weighted (relatively), or if it maintains the volume of normal text while increasing abnormal text.

**Justification Of The Preliminary Rating:**

The paper addresses a critical and well-recognized problem in medical report generation (class imbalance and normality bias) with a logical and effective solution. The rigorous evaluation on multiple datasets, including external validation, demonstrates clear clinical utility, particularly the significant improvement in F1-scores.

However, the method relies on established techniques (oversampling) applied to a structured sub-task, limiting its technical novelty. Additionally, the complexity of the multi-stage pipeline and the lack of analysis regarding inference costs and error propagation prevent a higher score. Nevertheless, the practical insights and the modular framework are valuable contributions to the MIDL community.

**Questions To Address In The Rebuttal:**

Inference Latency: Could you provide a comparison of the inference time/computational cost between the proposed modular ALO approach and the single-pass "Free-Text" or "Structured" baselines? Does the modularity introduce significant latency?

Cross-Anatomy Correlations: Did you observe any degradation in the coherence of the generated report regarding systemic diseases? For example, does the isolated training prevent the lung model from utilizing cues from the heart region?

Structuring Robustness: For the external AMOS-MM dataset, you used GPT-4.1 for structuring. Did you manually verify a subset of these to ensure the ground truth for your evaluation was accurate? How sensitive is ALO to noise in the structuring phase?

---

> ### Author Response · Authors · 2026-01-25
> **Response to Reviewer Pt1S**
>
> We thank Reviewer Pt1S for the positive and detailed assessment of our work, in particular for highlighting the effectiveness of ALO as a data-centric solution to class imbalance, the comprehensive evaluation across internal and external datasets, and the clarity of the presentation. We address the reviewer’s questions and concerns below.
>
>
> ### [1] *“Inference latency and computational overhead of ALO”*
> > We agree that inference latency is an important consideration. As discussed in the limitations (Section 5.3), the modular ALO pipeline introduces additional inference cost compared to single-pass baselines. In our experiments, ALO inference is approximately 1.6x slower than the Free-Text baseline at the dataset level. However, anatomy-level generation can be parallelized in practice, and we consider this overhead an acceptable trade-off for improved clinical sensitivity. For transparency, we have added the relative increase of the inference time in the limitations paragraph (Section 5.3).
>
>
> ### [2] *“Cross-Anatomy Correlations”*
> > Anatomy expert models generate findings independently but operate on shared image features extracted from the full CT volume, allowing each expert to access global visual context. Cross-anatomical consistency is further handled by the impression generation model, which jointly reasons over all anatomy-level findings. We agree that cross-anatomy dependencies are important and have added this design consideration in the revised manuscript (Section 3.3).
>
>
> ### [3] *“Structuring Robustness”*
> > For AMOS-MM, report structuring is used solely to ensure compatibility with the CT-RATE anatomy-level evaluation protocol and follows the procedure described in RadGenome-ChestCT (Zhang et al., 2024). We manually inspected a subset of structured reports (see Appendix, Fig. 7) and found them to be consistent with the original free-text findings. The sensitivity of ALO to structuring noise is difficult to quantify and remains an open research question, which we briefly discuss as future work in the revised manuscript limitations (Section 5.3).
>
> > Zhang, X., Wu, C., Zhao, Z., Lei, J., Zhang, Y., Wang, Y., & Xie, W. (2024). Radgenome-chest ct: A grounded vision-language dataset for chest ct analysis. arXiv preprint arXiv:2404.16754.
>
>
> ### *“Pipeline Complexity”*
> > We performed a quality check of the structuring stage by comparing the structured reports to the original ground-truth free-text reports using the same evaluation metrics as reported in Table 2. The consistently high scores (GREEN 0.959 or F1 0.964) indicate that the overall report content is preserved during structuring. We have added the corresponding results as a table in the appendix (A.4.). Evaluating the correctness of anatomy-level assignments themselves is more challenging and remains an important direction for future work.

---

### Official Review · Reviewer_syZj · 2026-01-10

**Confidence:** 4
**Preliminary Rating:** 2
**Final Rating:** 3

**Summary:**

This paper introduces Anatomy-Level Oversampling (ALO), a straightforward data-centric strategy that restructures reports into anatomy-level sections and balances the data distribution within each region. Experiments on three public datasets (CT-RATE, RAD-ChestCT, and AMOS-MM) demonstrate that ALO consistently improves classification scores while maintaining comparable language quality. The results suggest that for report generation, enhancing the supervision signal can be as impactful as architectural innovations.

**Strengths:**

* The proposed ALO framework is conceptually simple and can be easily integrated into existing report-generation pipelines.
* The paper compares the proposed model’s performance against baselines across multiple dimensions, including clinical metrics, natural language generation (NLG) quality, and classification performance. Furthermore, the authors provide thorough experiments and clear explanations of the results.

**Weaknesses:**

* While the ALO strategy is designed to address class imbalance in medical reports, its underlying logic bears similarities to Focal Loss, which makes the author's contribution distinct and non-trivial.
* Currently, the experiments do not include a direct comparison between the proposed ALO strategy and other standard training techniques for addressing class imbalance (e.g., weighted loss functions).
* Although ALO yields improvements in classification performance, these gains are come with trade-off in some clinical metrics.
* The proposed method rely on another tool for data labeling.

**Detailed Comments:**

Confidence intervals are missing from Table 2, which are essential for assessing performance. Because some scores show high variance, it is unclear whether the gains achieved by ALO are statistically significant.

**Justification Of Final Rating:**

I would like to thank the authors for thoroughly addressing my concern regarding the mechanism of oversampling and clarifying it in the manuscript. I am happy to increase the final rating accordingly.

**Justification Of The Preliminary Rating:**

The proposed ALO strategy addresses class imbalance via anatomical balancing; however, oversampling long-tail classes is a common approach. Additionally, because ALO acts as a standalone preprocessing step which means the model's gradients do not flow into the upstream components, limiting the framework's ability to learn an optimal sampling distribution dynamically.

**Questions To Address In The Rebuttal:**

Please address the questions in weakness and detailed comments.

---

> ### Author Response · Authors · 2026-01-25
> **Response to Reviewer syZj**
>
> We thank Reviewer syZj for the careful assessment of our work, in particular for highlighting the conceptual simplicity of ALO, its easy integration into existing report-generation pipelines, and the thorough multi-metric experimental evaluation. We address the reviewer’s concerns in the following points.
>
>
> ### [1], [2] *“Clarification of novelty and relation to standard class-imbalance techniques”*
> > We thank the reviewer for raising this important point regarding the relation to standard imbalance-handling methods. Our model is a large language model trained to generate free-text radiology reports, not to predict a single class label. Training is performed using token-wise cross-entropy over long textual outputs, and pathology labels are used only for post-hoc evaluation, not as direct supervision. Consequently, loss-based imbalance methods such as weighted loss or focal loss, which assume explicit class-level targets, are not directly applicable in this setting.
>
> > Moreover, class balancing in patient-level free-text report generation is not straightforward: naïve patient-level oversampling would also increase normal findings from unrelated anatomies, thereby reinforcing the normality bias. ALO addresses this by rebalancing abnormal content per anatomy, enabling targeted imbalance correction without amplifying normal findings elsewhere. To our knowledge, such anatomy-level balancing has not been explored in prior work on radiology report generation. We have clarified this distinction in Section 2 (Related Work) of the revised manuscript.
>
>
> ### [3] *“Trade-offs between classification gains and clinical metrics”*
> > We acknowledge that ALO introduces trade-offs in some clinical metrics. Prior work has shown that several standard clinical metrics tend to over-reward high-precision reports dominated by normal findings, which can lead to underreporting of abnormalities (Hamamci et al., 2025). In clinical practice, high sensitivity to pathologies requiring immediate attention (e.g., pulmonary nodules) is often more critical. Accordingly, we focus on Recall, F1-Score, and pathology-level evaluation, as emphasized in recent benchmarks such as the VLM3D challenge. As shown in Figures 3 and 4, ALO substantially improves recall across clinically relevant pathologies. For transparency, we still report the full set of clinical, NLG, and classification metrics. We added a short paragraph explaining the motivation for the highlighted metrics in Table 2 in the revised manuscript (Section 4.3).
>
> > Hamamci, I. E., Er, S., Shit, S., Reynaud, H., Kainz, B., & Menze, B. CRG Score: A Distribution-Aware Clinical Metric for Radiology Report Generation. In Medical Imaging with Deep Learning-Short Papers.
>
>
> ### [4] *“The proposed method rely on another tool for data labeling”*
> > We note that relying on existing tools for data labeling is common practice in the field. Many established methods build on external software or pretrained models (e.g., perceptual losses based on VGG networks). In this sense, our approach adheres to standard practice and leverages well-established tools as part of the overall methodology.
>
>
> ### [Detailed Comment] *“Confidence intervals are missing from Table 2”*
> > We thank the reviewer for this suggestion. As is common in prior report generation work, results are reported at the dataset level on the full official validation splits, as a single evaluation run is already time-consuming (e.g., evaluating GREEN on 3,000 samples takes over one hour). We therefore focus confidence interval analysis on classification-based metrics, which are central to our claims and to recent benchmarks such as VLM3D, and have added bootstrap-based confidence intervals for these metrics in the revised manuscript (Table 2).

---

### Author Rebuttal · Authors · 2026-01-25

**Rebuttal:**

We sincerely thank the reviewers for their insightful and constructive feedback. We appreciate the recognition of the conceptual simplicity of ALO, its easy integration into existing report-generation pipelines, its effectiveness as a data-centric solution to class imbalance, and the thorough experimental evaluation across multiple datasets. The reviewers’ comments have helped us improve the clarity and completeness of our work.

We upload a revised manuscript with the following changes:
- **Section 2 (Related Work):** clarified the distinction between ALO and standard class-imbalance techniques (e.g., weighted loss, focal loss) and discussed why these methods are not directly applicable to free-text report generation.
- **Section 3 (Methods):** expanded the description of the anatomy-level structuring and oversampling logic to improve technical transparency and reproducibility.
- **Section 3.3:** clarified how cross-anatomy consistency is handled via shared visual features and impression generation.
- **Section 4.3 and Table 2:** added bootstrap-based confidence intervals for classification metrics and clarified the motivation for emphasizing recall- and pathology-level evaluation.
- **Section 5.3 (Limitations):** added a discussion of inference latency, modular pipeline overhead, and sensitivity to structuring noise.
- **Appendix A.4:** added a quantitative quality check of the report structuring stage.
- **Figures 1 and 2:** labeled processing steps for improved clarity.
- **Minor edits:** corrected formatting issues (e.g., quotation marks) throughout the manuscript.

All changes in the revised manuscript are highlighted in red. We also provide detailed point-by-point responses to all reviewers’ comments in the official comments for each reviewer.

**Supporting Material:**

/attachment/30559378f0a0d4812714cd4a23209103e9d4d49b.pdf

---

### Meta-Review · Area_Chair_Zmbq · 2026-02-13

**Recommendation:** Accept (Poster)
**Confidence:** 4

**Metareview:**

In general, the ratings are mostly positive. I agree with the reviewers to accept this paper.

---

### Decision · Program_Chairs · 2026-02-14

Accept (Poster)